

# Enhancing nighttime cloud optical and microphysical properties retrieval using combined imager and sounder from geostationary satellite

5    Xinran Xia[1], Min Min[1],*, Jun Li[2], Yiming Zhao[3], Ling Gao[2], Bo Li[2]

[1] School of Atmospheric Sciences and Guangdong Province Key Laboratory for Climate Change and Natural Disaster Studies, Sun Yat-sen University and Southern Laboratory of Ocean Science and Engineering, Zhuhai 519082, China

10   [2] Innovation Centre for Fengyun Meteorological Satellite (FYSIC), National Satellite Meteorological Centre, China Meteorological Administration, Beijing 100081, China

[3] State Key Laboratory of Environment Characteristics and Effects for Near-space, Beijing Institute of Technology, Beijing 100081, China

*Correspondence to*: Min Min (minm5@mail.sysu.edu.cn)





**Abstract.** Accurate retrieval of cloud optical and microphysical properties (COMP) at night is important for monitoring changes in weather and climate systems. The nighttime cloud optical and microphysical properties (NCOMP) retrieval is enhanced by integrating data from hyperspectral infrared sounder and high-resolution imager on

the same geostationary platform with a machine learning framework. Using geostationary satellite imager broadband thermal infrared (TIR) channels along with dozens of optimally selected hyperspectral IR (HIR) channels, we demonstrate substantial improvements over traditional TIR-channel-based methods. The HIR channels enhance sensitivity to cloud effective radius (CER) and optical thickness

(COT), particularly for optically thin clouds, reducing retrieval errors to 9.73 μm and 6.09, respectively, with an approximate 10% accuracy improvement. The ML-based model preserves strong day-night continuity in COMP retrievals and assures the diurnal information for clouds, although challenges remain for thick clouds. This work highlights the importance of GEO-satellite-based HIR sounders, which provide

critical spectral information that complements imager data for cloud optical and microphysical property retrievals. Middle-wave IR (MWIR) channels significantly improve COT retrieval. The proposed fusion approach offers a flexible retrieval framework applicable to future geostationary satellite systems for enhancing the cloud property retrievals containing diurnal information.


**Keywords:** Geostationary Infrared Hyperspectral Sounder; Cloud Optical and Microphysical Properties; Deep Learning.



## 1 Introduction

Clouds constitute essential components of the Earth-atmosphere system, profoundly modulating both the planetary radiation budget and hydrological cycling (Ramanathan et al., 2007; Stevens and Bony, 2013; Arias, 2023). Cloud optical and microphysical properties (COMP), notably cloud optical thickness (COT) and cloud effective radius (CER), quantify two critical dimensions: COT represents the

integrated extinction coefficient of cloud particles, while CER characterizes the dominant particle size distribution (Nakajima and King, 1990; Liu et al., 2014; Wang et al., 2016). These parameters are pivotal for quantifying cloud radiative forcing mechanisms, as they govern the scattering and absorption efficiencies for solar shortwave and terrestrial longwave radiation, thereby directly regulating Earth's net

radiative equilibrium (Miller et al., 2018; Teng et al., 2023; Tana et al., 2023).

Satellite remote sensing has become the primary method for COMP retrieval due to its unique combination of large spatial coverage and high temporal resolution capabilities. The foundational work of Nakajima and King (1990) established the physical retrieval framework for deriving daytime COT and CER from solar reflective

channels (Nakajima and King, 1990). In this approach, the VIS/NIR channels (0.65/0.86 $\mu$m) primarily respond to COT with negligible cloud absorption, whereas the SWIR channels (1.61/2.13 $\mu$m) exhibit sensitivity to both COT and CER due to substantial cloud absorption (Platnick et al., 2003). This classical physically-based methodology has since become the standard for operational COMP products retrieved

by major satellite data application centers, including NASA Earth Observing System - Moderate Resolution Imaging Spectroradiometer (EOS-MODIS) and the EUMETSAT Spinning Enhanced Visible and InfraRed Imager (SEVIRI) (Salomonson et al., 2002; Thies et al., 2008), delivering global COMP such as Terra/Aqua MODIS for the past few decades (Platnick et al., 2017). Simultaneously,

current geostationary meteorological satellite systems, such as the GOES-R (Geostationary Operational Environmental Satellite - R series), Himawari-8/-9, and FY-4 (Fengyun-4) series, etc., offer 10/15-minute-scale COMP retrievals with 2-4 km spatial resolution, enabling new opportunities for investigating rapidly evolving cloud systems and their diurnal variations (Andi et al., 2013; Min et al., 2017; Letu et al.,

2022).



However, the aforementioned COMP algorithm is limited during the daytime (or called as DCOMP), as it relies on information from visible light scattering and near-infrared absorption. Without solar radiation at night, estimating the COMP becomes a challenging task (Gong et al., 2018). To address this issue, the split-window method has been proposed for retrieving COMP during the nighttime (Inoue, 1985; Heidinger and Pavolonis, 2009). This method, which relies solely on information from thermal infrared (TIR) channels, is applicable only to the detection of optically thin clouds (COT<5) at night due to the lack of spectral sensitivity of TIR channels to optically thick clouds (Heidinger and Pavolonis, 2009; Minnis et al., 2011; Iwabuchi et al., 2014). Compared to the operational DCOMP algorithm, the nighttime cloud optical and microphysical properties (NCOMP) algorithm produces estimates of COT that significantly lower than Daytime, makes it challenging to study the diurnal variation of cloud properties (Minnis and Heck, 2020; Y. Li et al., 2022). Recent advancements address this limitation by utilizing lunar reflectance from the Day/Night Band (DNB) and 3.9 μm channel emissivity. However, challenges persist in urban areas, where background city light signals may interfere with accurate retrievals (Walther et al., 2013; Min et al., 2021).

Recently, with the rise of artificial intelligent (AI) technology, which is capable of extracting complex relationships between multiple features, significant progress has been made in cloud properties retrieval (Minnis et al., 2016; Håkansson et al., 2018; Wieland et al., 2019; Min et al., 2020; Yang et al., 2022). By constructing the relationship between thermal infrared (TIR) channels and daytime cloud optical and microphysical properties, NCOMP can be estimated using the TIR channels. Minnis et al., (2016) developed a neural network algorithm to estimate the nocturnal COT of opaque ice clouds using 3.7, 6.7, 11.0, and 12.0 μm infrared channels. The algorithm shows a high degree of consistency in validation results and extends the ice cloud COT estimate range to 150 (Minnis et al., 2016). Wang et al. (2022) introduced a convolutional neural network (CNN) model to retrieve COT, CER, and cloud top height (CTH). Based on MODIS TIR channels, this method extends COT and CER retrievals into the nighttime and outperforms the traditional TIR -based method, the validation results show a good correlation for CTH (R = 0.95), CER (R = 0.85) and COT (R = 0.79) (Wang et al., 2022). Zhao et al., (2023) developed a new ResUnet-model-based algorithm for retrieving cloud phase (CLP), COT, CER, and



CTH. With geostationary satellite imager multichannel brightness temperature (BT) data to input, this model provides reliable and comparable COMP estimates within the COT range of 0-60, the RMSEs (root-mean-square error) of the CER and COT are 7.14 μm and 9.01, with higher accuracy at values under 20 (Zhao et al., 2023). Furthermore, Charles et al. (2024) trained multiple neural network models for emulating CER and COT using only TIR channels from the GOES-16 ABI (Advanced Baseline Imager). This ML-based approach significantly improves NCOMP estimation compared to the operational NCOMP product and particularly reduces artifacts associated with the day/night terminator.

Hyperspectral infrared (HIR) sounders, such as the Infrared Atmospheric Sounding Interferometer (IASI), the Atmospheric Infrared Sounder (AIRS), and the Cross-track Infrared Sounder (CrIS), offers significant advantages in retrieving the vertical structure of atmospheric temperature, humidity, and trace gases (Menzel et al., 2018). The high spectral resolution of HIR sounders also makes them well suited to the retrieval of cloud properties (Li et al., 2004, 2005). Through radiative transfer simulations, Huang et al., (2004) demonstrated that the IR spectrum between 790-960 cm$^{-1}$ is sensitive to the CER, while the 1050-1250 cm$^{-1}$ range is sensitive to the COT. Based on these spectral features, a method for retrieving COMP from ice clouds using HIR data was proposed, which applied one-dimensional variational (1DVAR) and minimum-residual (MR) methods to retrieve COMP from the AIRS longwave window region (Li et al., 2005). In addition, Liu et al., (2009) also converted IASI channel radiance spectra into super-channels and used a Principal Component-based Radiative Transfer Model (PCRTM) to retrieve atmospheric temperature, moisture, and cloud optical properties (Liu et al., 2009).

The Geostationary Interferometric Infrared Sounder (GIIRS) - the first hyperspectral infrared (HIR) sounder on a geostationary platform, provides significantly higher temporal resolution for vertical atmospheric profiling than polar-orbiting hyperspectral instruments, enabling real-time monitoring of rapidly evolving weather systems on a regional scale (Yang et al., 2017; Li et al., 2022; Kalluri, 2022; Li et al., 2025). This provides a valuable opportunity to further explore the benefits of combining both HIR sounders and imagers onboard a GEO satellite for retrieving COMP. Guo et al., (2024) developed a cloud macro-physical properties retrieval algorithm by integrating data from the Advanced Geosynchronous Radiation Imager (AGRI) and GIIRS. Their results demonstrate that GIIRS significantly





improves the accuracy of cloud phase, CTH, and cloud base height (CBH) retrievals, marking the first validation of HIR's capability for probing cloud vertical structures (Guo et al., 2024). With Europe, Japan, and the United States planning next-generation GEO HIR deployments (Bessho et al., 2021; Holmlund et al., 2021), operational applications of GEO HIR data (particularly for diurnal cloud/wind field products) have become a critical research frontier in atmospheric science (Lindsey et al., 2024).

Therefore, the primary goal of this study is to answer the following two key scientific questions: (1) What is the advantage of a GEO HIR sounder over a GEO IR imager for NCOMP retrieval? (2) How does a combined GEO satellite imager/sounder improve current imager-TIR-channel-based NCOMP algorithms? To answer these two scientific questions, we propose a new NCOMP retrieval framework that utilizes a combined IR imager and HIR sounder onboard a GEO FY-4B satellite platform and machine-learning-based algorithm. The specific effect of FY-4B HIR observations in NCOMP retrieval is quantitatively analyzed and evaluated. The subsequent sections are organized as follows: Section 2 provides a comprehensive description of the data and methodology employed; Section 3 evaluates the contribution of GIIRS channels to COMP retrieval and validates the accuracy of the COMP retrieval algorithm; Section 4 presents the conclusion and discussion.

## 2 Data and Method

### 2.1 Data Collection

AGRI is one of the key sensors on board the FY-4B geostationary meteorological satellite, which was successfully launched in June 2021. AGRI provided a full disk image of the Eastern Hemisphere every 15 minutes, with a total of 15 channels: visible/near infrared (VIS/IR) channels (0.47 - 0.825 μm), shortwave infrared (SWIR) channels (1.379 - 2.25 μm) and TIR channels (3.75 - 13.3 μm). Of these, the reflectance observations provided by 0.64 μm and 2.25 μm channels are mainly used to generate the DCOMP product. Since the VIS/IR channels are unavailable during nighttime and to minimize the effects of sun glint over ocean surfaces, we selected the TIR channels (channels 9–15) along with the satellite viewing zenith angle (VZA) as input features for developing the NCOMP retrieval



algorithm. Additionally, the AGRI Level-2 (L2) cloud products (CLP, CER, and COT) were employed as target outputs for the model (Wang et al., 2024a).

GIIRS is the first hyperspectral IR sounder on board a geostationary satellite. Compared to its predecessor FY-4A/GIIRS, FY-4B/GIIRS improves the spatial resolution from 16 km to 12 km at nadir and extends the spectral range in longwave infrared (LWIR) channels (from 700-1130 cm$^{-1}$ to 680-1130 cm$^{-1}$) (Di et al., 2021). Besides, evaluations show that FY-4B/GIIRS also enhances calibration stability and accuracy compared to FY-4A/GIIRS, particularly in reducing errors within $CO_2$ and $O_3$ absorption channels (Niu et al., 2023; Wang et al., 2024b). In this study, we quantitatively evaluate the sensitivity of GIIRS LWIR and MWIR to COMP and further explore the hyperspectral IR sounder gain in NCOMP retrieval. Note that on March 4, 2024, FY-4B underwent an orbital drift, adjusting its orbital nadir position from 133°E to 105°E, to succeed FY-4A and enhance its operational monitoring services. In this study, data from March and July 2024 are utilized for ML model training, while five days of data from March, June, and July have been selected for independent model validation. Table 1 summarizes the relevant variables used in ML model training and prediction.

### 2.2 Data Prepossessing

To maintain spatiotemporal consistency between input features and targets, we collocated each GIIRS pixel with multiple simultaneous AGRI observations, with the final output resolution is consistent with AGRI (4 km). The criterion of matching is that the distance is within 4 km and the temporal difference is less than 15 minutes (Di et al., 2023). FY-4B/GIIRS performs sequential scanning of individual granules from top to bottom at 15-minute intervals, completing a full scan of the China region through 7 granules per scanning cycle. Meanwhile, FY-4B/AGRI conducts full-disk scans every 15 minutes. Therefore, in each scanning cycle, the synchronized matching data from GIIRS and AGRI are also divided into 7 independent granules. We implemented comprehensive quality control for the matched dataset: (1) exclusion of observations with solar zenith angles larger than 65° (limiting analysis to daytime COMP retrievals), and (2) removal of data with viewing zenith angles larger than 65° (Andi et al., 2013). To speed up model training while reducing the impact of outliers on model training, we also used the z-score method to normalize the data,



transforming the model input data into a standard normal distribution with mean 0 and standard deviation 1. During ML-based model training, all input data were partitioned into 32×32 pixels regions (corresponding to 128 km×128 km). This processing yielded 256, 8230 samples for training and 49, 2153 samples for testing and
validation.

### 2.3 Machine-Learning-Based Algorithm

ML models based on the advanced Unet architecture have become prevalent in medical imaging, remote sensing, and natural image processing (Ronneberger et al., 2015). In recent years, researchers have made various improvements based on Unet,
to further enhance its performance in complex scenes (Zhou et al., 2018; Diakogiannis et al., 2019). Sun et al., (2023) proposed the DA-TransUnet (Dual-Attention Transformer Unet) architecture, which incorporates dual-attention mechanisms for positional and channel information processing, achieving improved segmentation efficiency without compromising performance (Sun et al., 2023).

### 2.3.1 Model Architecture

Inspiring from Sun et al., (2023) we developed an architecture for FY-4B/AGRI COMP retrieval, named HIR-COMP-Unet, the detailed model architecture is shown in Figure 1. The model is mainly composed of five parts: input layer, encoder layer, decoder layer, CLP enhanced layer, and output layer. In the input
layer, AGRI BTs, VZA (view zenith angle), and GIIRS BTs are loaded with a size of 32×32 in a batch, considering the spatial resolution of the GIIRS BTs is coarser than AGRI BTs, the gaussian filter is used to smooth the GIIRS BTs. The encoder layer of the model consists of multiple convolutional blocks, each followed by a dual attention (DA) block and a 2×2 max-pooling layer to progressively downsample the feature
maps while extracting positional and channel features. The convolutional blocks, improved by the ConvNext architecture, are designed to enhance feature extraction efficiency, while the DA blocks excel in both position-based and channel-based feature extraction, ensuring robust feature learning. In the bottleneck of the encoder is a Transformer block, which employs self-attention mechanisms to capture long-range
dependencies and global contextual information, further enriching the feature representations. The decoder part mirrors the encoder structure but uses upsampling layers instead of pooling to gradually restore the spatial resolution of the feature maps.



Each decoder layer combines upsampled features with skip connections from the corresponding encoder layer, enabling the model to retain fine-grained spatial details. After decoder, the initial classification and regression results are output with a size of Batchsize(B)×7×32×32, while 5 channels represent the predicted probabilities of the five cloud phase classifications, the other 2 channels represent the initial predictions of CER and COT. The CLP feature maps are processed by convolutional layers and batch normalization, followed by ReLU activation, to enhance the regression prediction result of CER and COT. Finally, the 5 channels classification predicted probabilities are softmax to CLP, the other 2 channels are CER and COT, respectively.

### 2.3.2 Loss Function

The loss function is used to quantify the difference between the model′s predictions and the targets, and the model gradually adjusts the parameters to become optimal as it minimizes the loss function. As the model performs both regression and classification tasks, the overall loss function is defined as follows:

$$L(Total) = L(\text{CLP}) + L(\text{CER}) + L(\text{COT}), \qquad (1)$$

where $L$ represents the loss function, the total loss function is composed of the loss functions of CLP, CER and COT. The Mean Squared Error (MSE) is used to calculate the loss functions of CER and COT, while the Binary Cross-Entropy (BCE) is used to calculate the loss function of CLP.

### 2.3.3 Training Strategies

Each ML-based model is trained in a batch of size B×C×32×32, where B and C represent the batch size and channel number, respectively, and 32×32 represents the pixel size of each image. We choose a batch size of 64 to capture as many local COMP features as possible. The channel number depends on the number of GIIRS channels we choose to add, a control experiment is set up to test the accuracy of COMP retrieval with different numbers of GIIRS channels to add, which mainly includes the following four scenarios: no GIIRS added (AGRI only), limit GIIRS LWIR, limit GIIRS MWIR, and both GIIRS LWIR and LWIR. To improve the robustness of the ML model, the sample augmentation strategy is used during the training process, which means in each batch of model training, the training images are randomly rotated by 90°, 180° or 270° as input.



To facilitate rapid convergence during model training and to mitigate the risk of gradient explosion, we implemented several optimization strategies, including gradient clipping and adaptive learning rate scheduling. With an initial learning rate of 1.0e-3, the scheduler reduces the learning rate by a factor of 0.5 when no improvement in validation loss is observed for 10 consecutive epochs, with a
minimum learning rate set at 1.0e-6. Each model is trained on a platform with an Intel Xeon(R) Gold 5318Y CPU (2.10 GHz, 96 cores) and two NVIDIA GeForce RTX 4090 GPUs with 300 epochs.

## 3 Results and Validation

### 3.1 Sensitivities of GIIRS radiances to COMP retrieval

Given that the FY-4B/GIIRS comprises over 1600 spectral channels, direct incorporation of all channels into ML-based COMP retrieval models would reduce computational efficiency and may be counterproductive. Therefore, a systematic channel selection approach is essential to identify: (1) channels exhibiting optimal sensitivity to COMP, and (2) channels demonstrating minimal radiometric calibration
errors. This selective utilization of hyperspectral data ensures computational efficiency while maintaining the physical interpretability of the ML-based retrieval system.

    The Radiative Transfer for TOVS (RTTOV) model, developed by the European Centre for Medium-Range Weather Forecasts (ECMWF), offers significant
advantages for hyperspectral infrared simulations due to its computational efficiency and well-validated gas absorption parameterization schemes (Saunders et al., 2018). In this work, we employ RTTOV version 13.2 to systematically evaluate the sensitivity of GIIRS LWIR and MWIR spectral channels to COMP. Our simulation framework examines two fundamental cloud scenarios: liquid-phase and ice-phase
clouds, a controlled experiment is designed where we systematically vary either CER or COT while holding other cloud parameters (cloud height, cloud top pressure and cloud fraction) constant. The quantitative relationship or sensitivity (Sen) between GIIRS radiance and COMP variations is established through the following equation:

$$\text{Sen} = \left| \frac{\Delta BT/BT}{\Delta COMP/COMP} \right|, \tag{2}$$



where ΔCOMP represents variations in CER and COT. The symbol of ΔBT
denotes the corresponding changes in GIIRS brightness temperature (BT) due to these
variations.

For liquid water clouds, Figure 2 reveals distinct spectral sensitivity
characteristics to CER when COT is fixed at 1. The LWIR channels (750-1000 cm$^{-1}$

and 1080-1130 cm$^{-1}$) demonstrate a strong sensitivity of approximately 0.4 -to droplet
sizes of 2-20 μm, while MWIR channels (1800-2200 cm$^{-1}$) exhibit a moderate
sensitivity of approximately 0.2 for 4-20 μm droplets. When analyzing COT
sensitivity with CER fixed at 10 μm (Figure S1), the LWIR channels demonstrate a
strong sensitivity of approximately 0.7, particularly for COT values below 10.

Although MWIR channels show relatively weaker response to COT variations
compared to LWIR, they retain significant sensitivity (about 0.5) in the 1800-2200
cm$^{-1}$ range for optically thin clouds.

For ice clouds, the spectral sensitivity to CER displays different patterns (Figure
S2). The LWIR channels (900-1000 cm$^{-1}$ and 1080-1130 cm$^{-1}$) maintain a strong

sensitivity of approximately 0.4 to particle sizes of 10-20 μm, while MWIR channels
show minimal response to CER variations overall. In COT sensitivity analysis
(CER=20 μm), LWIR channels (750-1000 cm$^{-1}$ and 1080-1130 cm$^{-1}$) again
demonstrate a strong sensitivity of approximately 0.6 for COT values below 3, with
MWIR channels exhibiting reduced but measurable sensitivity in the 1950-2200 cm$^{-1}$

range (Figure S3).

We further calculate the average sensitivity of the GIIRS LWIR and MWIR
channels to liquid-phase and ice-phase clouds CER and COT under RTTOV
simulations (Figure 3(a), (b)). The LWIR channels exhibit a strong sensitivity of
approximately -0.3 to COT variations in spectral band of range 750-950 cm$^{-1}$ and

1080-1130 cm$^{-1}$, while showing a maximum but relatively weak CER sensitivity of
approximately -0.15 in the 900-1000 cm$^{-1}$ range. Following the previous work by
Wang et al., (2024), the radiometric calibration performance of FY-4B/GIIRS was
evaluated using IASI observations and RTTOV simulations as benchmarks (Figure
3(c), (d)). The results demonstrate that GIIRS exhibits greater sensitivity to COT than

to CER, with the most responsive spectral channels located at 700-1000 cm$^{-1}$ and
1080-1130 cm$^{-1}$ in the LWIR region and 2100-2170 cm$^{-1}$ in the MWIR region.
Comparative analysis reveals that in the LWIR channels, GIIRS maintains a mean



bias within 0.5 K relative to IASI with a standard deviation (STD) of approximately 1.5 K, while showing maximum COMP sensitivity in the 700-1000 cm⁻¹ and 1080-1130 cm⁻¹ channels. In contrast, MWIR channel performance is comparatively weaker, with errors reaching 2 K in the 1650-1850 cm⁻¹ and 2200-2250 cm⁻¹ ranges, the latter also showing a higher STD (Wang et al., 2024b).

In order to identify high-quality GIIRS channels with optimal sensitivity to COMP, we established the following selection criteria: (1) the median COMP sensitivity for each selected channel must exceed 0.2; (2) the STD of calibration uncertainty must be less than 1.5 K; and (3) the absolute bias must remain below 0.5 K. Based on these stringent requirements, we ultimately selected 149 LWIR channels and 40 MWIR channels. As shown in Figure 4(a), the chosen LWIR channels are predominantly clustered in the 720-900 cm⁻¹ and 1000-1060 cm⁻¹ spectral channels, while the MWIR channels are primarily concentrated in the 2100-2180 cm⁻¹ band. Analysis of the temperature and water vapor mixing ratio Jacobians for these selected channels (Figure 4(b), (c)) reveals that they contain valuable temperature and water vapor information across different atmospheric layers, which provides theoretical basis for subsequent COMP retrieval. The selected GIIRS channels were prioritized based on their Principal Component Analysis (PCA) scores, determined through the following formula:

$$\text{Importance}_j = \sum_{i=1}^{k}(v_{ij}\sqrt{\lambda_i})^2, \tag{3}$$

where k denotes the number of retained principal components, $v_{ij}$ represents the eigenvector coefficient of the j-th channel for the i-th principal component, $\lambda_i$ represents the eigenvalue associated with the i-th principal component. We ranked all selected LWIR and MWIR channels according to their importance scores. In COMP retrieval control experiments with different numbers of GIIRS channels input, channels with high importance scores are prioritized.

### 3.2 COMP retrieval and comparisons

To investigate the role of HIR in COMP retrieval and determine the optimal observation channel scheme for HIR-COMP-Unet, we trained models using four primary channel selection strategies: (1) AGRI IR only, (2) AGRI IR + GIIRS LWIR, (3) AGRI IR + GIIRS MWIR, and (4) AGRI IR + both GIIRS LWIR and MWIR (The specific channel selection is shown in Figure S4). Additionally, to further evaluate the influence of the number of GIIRS input channels on COMP retrieval accuracy, we



conducted a tiered experiment with varying numbers of input channels, prioritizing those with the highest importance.

Table 2 presents the COMP retrieval accuracy across different channel schemes using FY-4B/AGRI L2 daytime COMP product as reference (Wang et al., 2024a). For CER retrieval, RMSE, MAE and MBE of 12.85 μm, 8.32 μm and -3.30 μm as only using 7 FY-4B/AGRI TIR channels. While adding 10 LWIR channels initially increased the RMSE to 13.36 μm, further channel inclusion (up to 50 channels) improved the accuracy to 12.74 μm. In contrast, incorporating MWIR channels negatively impacted the CER retrieval performance. For COT retrieval, using AGRI TIR channels alone resulted in RMSE, MAE, and MBE values of 8.96, 4.55, and -0.89, respectively. Introducing LWIR channels initially raised the RMSE to 9.87 with 10 channels, but subsequently reduced it to 9.07 with 50 channels. Conversely, MWIR channels significantly improved the COT retrieval, reducing the RMSE to 7.71 when using 30 channels. An optimal balance was achieved by combining 30 LWIR and 10 MWIR channels, yielding an RMSE of 13.13 μm for CER and 7.84 for COT, thus enhancing COT retrieval while maintaining CER accuracy.

FY-4B/AGRI L2 daytime COMP product from June 2024 is used for independent validation, focusing on cases with CER below 60 μm and COT below 60 (covering 99.3% of samples). Using only AGRI IR channels (Figure 5), HIR-COMP-Unet achieved cloud classification accuracies of 73.98% (clear), 89.09% (water), 94.28% (supercooled), 87.80% (mixed), and 97.05% (ice). The model showed relative difficulty distinguishing clear sky from water clouds (26.01% misclassification) while excelling in ice-phase cloud identification (>94% accuracy). For CER retrieval, the RMSE, MAE and MBE are 9.72 μm, 7.20 μm and -2.05 μm respectively, with most cases in the 10-40 μm range. The probability density function (PDF) (Figure 5(c)) analysis shows an overestimation of the lower CER values and an underestimation of the higher values, increased probabilities for 28-34 μm and decreased probabilities for 36-60 μm compared to AGRI L2 cloud product. Compared to CER, the retrieval of COT demonstrated better agreement, with RMSE, MAE, and MBE values of 6.71, 4.14, and -0.46, respectively. Although the PDF distribution (Figure 5(e)) indicated slight deviations for optically thin clouds within the ranges of 0-2 and 4-6, the overall agreement with the reference data remained good.

For comparison, Figure 6 evaluates the HIR-COMP-Unet model performance using 30 LWIR + 10 MWIR GIIRS channels as additional inputs. The inclusion of





GIIRS data leads to slight but consistent improvements: clear-sky detection accuracy
increases to 74.85%, water cloud identification reaches 89.10%, while ice-phase cloud
accuracy remains high (>94%). For CER retrieval, the model maintains similar
precision to AGRI-IR-only inputs (RMSE=9.73 µm, MAE=7.20 µm) but with reduced
bias (MBE improves from -2.05 to -0.91). PDF distributions (Figure 6(c)) confirm

better agreement with AGRI L2 products, indicating mitigated systematic errors in
CER estimation. More significantly, COT retrieval shows marked improvement, with
error metrics decreasing by -10% (RMSE = 6.09, MAE = 3.62, MBE = -0.62) and
PDF analysis (Figure 6(e)) demonstrating excellent alignment with reference data.
The comparison was also performed with the VIIRS L2 COMP product as a reference,

and the results also confirmed the contribution of the GIIRS LWIR and MWIR
channels for improving COMP retrieval accuracy (The comparison is shown in Figure
S5, S6).

To quantify the benefits of GIIRS for COMP retrieval, Figure 7 evaluates
HIR-COMP-Unet model performance across different COT regimes using various

GIIRS channel combinations. For thin clouds (0.1 ≤ COT < 1), the CER retrieval
RMSE ranged from 12 to 20 µm, with GIIRS channels providing slight improvement
by approximately 0.5 µm, while COT retrieval demonstrates more significant
enhancement (RMSE decreasing from 1.0-2.2 to 0.5-1.7). This improvement reflects
the enhanced sensitivity of GIIRS LWIR/MWIR channels to optically thin clouds,

consistent with conclusions from Section 3.1. For medium clouds (1 ≤ COT < 10),
CER errors decrease naturally from 25 µm to 8 µm with increasing COT, showing
limited response to GIIRS inputs. However, GIIRS LWIR channels reduce COT
retrieval accuracy (RMSE increasing by about 1), although the combined
LWIR+MWIR configuration yields modest improvements for COT 7-10. For thick

clouds (10 ≤ COT < 100), GIIRS LWIR channels improve CER retrieval (reducing
RMSE by 1.0-1.5 µm) but exhibit minimal positive effect on COT retrieval, where
errors exceed 30 for COT > 60 due to TIR channel sensitivity constraints.

Figure 8 presents the retrieval performance across different cloud droplet size
categories. For small droplets (5 ≤ CER < 20 µm), the baseline CER retrieval

RMSE of 8-10 µm improves to 7-9 µm with GIIRS channel inputs. The COT retrieval
shows comparable enhancement, with MWIR channels reducing RMSE by 1-2. For





medium droplets (20 ≤ CER < 50 μm) and large droplets (50 ≤ CER < 100 μm), GIIRS channels do not significantly improve CER retrieval accuracy. Instead, MWIR channels increases the error in CER retrieval. However, for COT retrieval, the incorporation of MWIR channels reduces RMSE by 1-2, demonstrating their positive impact on COT estimation for medium and large cloud droplets.

In conclusion, the addition of GIIRS LWIR and MWIR channels enhances the accuracy of the COMP retrieval. For COT retrieval, GIIRS channels mainly improve accuracy for optically thin clouds with COT values less than 10, with MWIR channels significantly enhancing retrieval for large grain sizes, especially ice clouds. In contrast, for CER retrieval, GIIRS channels show only a slight improvement for optically thin clouds, while MWIR channels tend to increase retrieval errors for medium to large cloud particles. Overall, combining both LWIR and MWIR channels proves most effective, leading to a modest improvement in CER retrieval for small particle sizes and a 10% overall increase in COT retrieval accuracy. The HIR-COMP-Unet model, using 30 LWIR and 10 MWIR GIIRS channels, was ultimately chosen as the optimal model, which also proved the importance of both LWIR and MWIR in COMP retrieval.

### 3.3 Case analysis

As an image-based cloud retrieval approach, this study evaluates the HIR-COMP-Unet's capability to reproduce spatial patterns of cloud attributes through regional case studies. Figure 9 presents a comprehensive comparison of satellite observations, L2 cloud products, and HIR-COMP-Unet model retrievals for a target region (99°E-132°E, 20°N-40°N) on 30 June 2024 at 03:15 UTC. The 0.64 μm visible channel reflectance, crucial for DCOMP retrieval, effectively captures cloud spatial distribution, while the 8.55 μm and 13.3 μm thermal infrared channels, exhibiting reduced BTs in cloudy areas, primarily provide cloud-top information. The analyzed scene features a prominent southwest-northeast oriented cirrus system with surrounding thin cumulus clouds. HIR-COMP-Unet model demonstrates exceptional performance in cloud phase classification, accurately distinguishing between liquid, ice, and mixed-phase clouds while correctly identifying clear-sky regions. Regarding CER retrieval, the model shows particular strength for high-level clouds, producing reliable estimates in the 20-40 μm range. The retrieved cloud textures exhibit strong correspondence with the derived ice water path (IWP) and liquid water path (LWP)





distributions. However, the model tends to underestimate CER values for large droplets in low-level water clouds and supercooled clouds. Comparison with L2 products reveals generally high agreement, with HIR-COMP-Unet model showing improved COT estimation for optically thick clouds (COT > 60). Nevertheless, the model displays a systematic underestimation in regions where COT exceeds 70,

suggesting potential limitations of ML-based algorithm in retrieving extremely thick cloud conditions.

        Figure 10 illustrates a representative case of complex cloud distribution within a smaller area (92°E-100°E, 6°N-11°N), featuring coexisting cirrus, cumulus, and mixed-phase clouds. The visible channel observations reveal intricate textural patterns

in the cloud system, while the thermal infrared channels partially resolve cloud-top features but lack detailed structural information. The scene exhibits a punctate distribution of pure ice and water clouds, interspersed with regions of transitional cloud phases. HIR-COMP-Unet model demonstrates strong performance in cloud phase classification, particularly in distinguishing ice and mixed-phase clouds.

However, the algorithm shows minor limitations in detecting small clear-sky regions spanning only a few pixels. Regarding COMP, the retrieved CER and COT distributions show good overall consistency with the IWP and LWP edge contours. The CER values are predominantly clustered in the 30-40 μm range, where the model provides accurate estimates, though some structural details are lost in central portions

of the scene. Notably, the model tends to overestimate CER values (5-10 μm range) for mixed-phase clouds containing small droplets. For COT retrieval, HIR-COMP-Unet model achieves high spatial correspondence with reference data, effectively capturing fine-scale features within the 0-20 optical thickness range. The model successfully reproduces the spatial variability of cloud optical properties while

maintaining physically realistic distributions.

### 3.4 Nocturnal features of COMP

        Traditional DCOMP retrieval algorithms face limitations at night due to the absence of visible light observations, hindering the capture of complete diurnal variations in cloud optical and microphysical properties (COMP). In contrast, thermal

infrared (TIR)-channel-based COMP retrieval methods are unaffected by solar illumination conditions, effectively compensating for this shortcoming of VIS/NIR-dependent approaches. To further assess the ability of the HIR-COMP-Unet



model developed in this study to capture diurnal COMP variations, we analyzed a case study region (110°E-115°E, 25°N-30°N) within the GIIRS scanning range (Figure 10 (a)). At 11:00 UTC on 29 June 2024, the unavailability of DCOMP products prevented validation against reference cloud phase (CLP) data. However, operational CLP and cloud-top height references indicated a cloud system primarily composed of extensive cirrus, with localized cumulus in the northwest. Despite a minor misclassification of clear-sky pixels as liquid clouds in the northwest, the HIR-COMP-Unet model accurately reproduced the spatial distribution of cloud phases. For both CER and COT retrievals, the model produced consistent edge textures across various CLP and CTH regimes, indicating that AGRI and GIIRS TIR observations alone provide sufficient data for the HIR-COMP-Unet model to characterize COMP distributions, both day and night, without the need for additional CLP/CTH inputs.

To further validate the nocturnal COMP retrievals from HIR-COMP-Unet model, we compared its hourly outputs with ERA5 reanalysis cloud water (CLW) data by applying the COT-CER-CLW relationship from Minnis (1998) (showing in the supplementary materials). Figure 12 illustrates the diurnal variations of CLW, CER, and COT on 29 June 2024, highlighting clear daytime and nighttime trends. During the day (local time), both CER and COT slightly decreased, reflecting reduced CLW, with HIR-COMP-Unet model retrievals closely matching reference values. At night, CER exhibited a characteristic dip-and-recovery pattern, while COT remained stable, together driving CLW variations that were consistently validated by ERA5 data. This confirms the model's ability to accurately capture diurnal COMP cycles.

## 4 Conclusion and Discussion

This study introduces a novel approach to enhance COMP retrieval using hyperspectral infrared channel observations. We develop a ML-based model, HIR-COMP-Unet model, incorporating FY-4B/AGRI TIR channels alongside varying numbers of FY-4B/GIIRS LWIR and MWIR channels, selected for their sensitivity to COMP variations. The impact of different GIIRS spectral channel combinations on COMP retrieval is specifically evaluated, while also addressing the two questions raised at the beginning of this work. Key findings are summarized below:



(1) GIIRS LWIR (720-900 cm⁻¹ and 1000-1060 cm⁻¹) and MWIR (2100-2180
cm⁻¹) channels in TIR-channel-based NCOMP algorithms are sensitive to variations
in CER and COT in both liquid and ice-phase clouds. This sensitivity offers additional
information that improves COMP retrieval, especially for optically thin clouds.

(2) Combining GIIRS and AGRI enhances NCOMP retrieval accuracy compared
to AGRI-only retrievals. For COT retrieval, GIIRS channels mainly improve accuracy
for optically thin clouds (COT < 10). For CER retrieval, the improvements are
marginal for similar clouds. The best performance is achieved using 30 LWIR and 10
MWIR GIIRS channels, reducing RMSE to 9.73 μm for CER and 6.09 for COT, with
an approximate 10% improvement in COT retrieval accuracy.

(3) HIR-COMP-Unet model efficiently integrates the features of each input
channel, providing spatially consistent retrievals compared to reference data. However,
challenges remain for higher CER (CER > 60 μm) and COT (COT > 60) cases. The
model also maintains strong continuity between retrieved nighttime and daytime
COMP, allowing for reliable monitoring of diurnal COMP variations at regional
scales.

In summary, this study offers a comprehensive evaluation of HIR's role in COMP
retrieval, providing valuable insights for the development of future GEO HIR
missions. Unlike previous approaches that rely on HIR LWIR window channels, we
carefully select GIIRS LWIR and MWIR channels sensitive to COMP variations,
showing that MWIR observations significantly improve COT retrieval accuracy. The
retrieval error for COT is significantly lower compared to only TIR-channel-based
inversions, highlighting the added value of hyperspectral data for characterizing cloud
microphysical and optical properties. Additionally, if the radiometric calibration
performance of FY-4B/GIIRS improves further, the inversion accuracy is also likely
to increase. The fusion of GEO imagers and HIR sounders in our proposed method
not only enhances FY-4B satellite COMP retrieval but also provides a scalable
framework applicable to other GEO platforms with HIR sounders (e.g.,
next-generation GEO satellites or successors of GOES-R, MTG).

*Data availability.* The authors would like to sincerely acknowledge NOAA and
ECMWF for freely providing gfs (ftp://nomads.ncdc.noaa.gov/GFS/Grid4) and ERA5
(https://cds.climate.copernicus.eu/datasets/reanalysis-era5-single-levels?tab=downloa
d) data online for retrieval and comparison.



*Author contributions.* MM proposed the essential research idea. XX, MM and JL performed the analysis and drafted the manuscript. YZ LG and BL provided useful comments. All the authors contributed to the interpretation and discussion of results and the revision of the manuscript.


*Competing interests.* The authors declare that they have no conflict of interest.

*Acknowledgements.* The authors also would like to thank Facebook Inc. for freely
providing the Pytorch software (https://pytorch.org) online. This research was supported by the National Natural Science Foundation of China (Grants No. U2342201, U2142201, and 42175086), the Innovation Group Project of the Southern Marine Science and Engineering Guangdong Laboratory (Zhuhai) (Grant No. SML2023SP208), and the Science and Technology Planning Project of Guangdong
Province (Grant No. 2023B1212060019). We also acknowledge the high-performance computing support from School of Atmospheric Science of Sun Yat-sen University. Lastly, the authors sincerely thank the editor and anonymous reviewers for their insightful suggestions and constructive comments, which significantly improved the quality of this manuscript.





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




## Tables and Figures

**Table 1.** Data descriptions

| Sensor | Variable | Description | Spatial/Temporal Resolution |
|---|---|---|---|
| AGRI | Channel 9 | Central wavelength 6.25 μm Infrared | 4km/15min |
| | Channel 10 | Central wavelength 6.95 μm Infrared | 4km/15min |
| | Channel 11 | Central wavelength 7.42 μm Infrared | 4km/15min |
| | Channel 12 | Central wavelength 8.55 μm Infrared | 4km/15min |
| | Channel 13 | Central wavelength 10.80 μm Infrared | 4km/15min |
| | Channel 14 | Central wavelength 12.00 μm Infrared | 4km/15min |
| | Channel 15 | Central wavelength 13.3 μm Infrared | 4km/15min |
| | VZA | Observational zenith angle | 4km/15min |
| | L2 CLP | AGRI cloud phase product | 4km/15min |
| | L2 CER | AGRI cloud effective radius product | 4km/15min |
| | L2 COT | AGRI cloud optical thickness product | 4km/15min |
| GIIRS | LWIR Channels | Long-wave Infrared channels (680-1130 cm$^{-1}$) | 12km/1.5h (China region) |
| | MWIR Channels | Mid-wave Infrared channels (1650-2250 cm$^{-1}$) | 12km/1.5h (China region) |




**Table 2.** Comparison of COMP retrieval accuracy with different numbers of GIIRS channels are added. The configurations include: IR-only (using only AGRI infrared channels without GIIRS inputs), LWnum (using only num selected LWIR channels from GIIRS), MWnum (using only num selected MWIR channels from GIIRS), and LWnum1MWnum2 (combined use of num1 LWIR and num2 MWIR channels from

GIIRS).

| Channels Input | Evaluation Indicators | IR-only | LW10 | LW30 | LW50 | MW10 | MW30 | LW30 MW10 | LW50 MW10 |
|---|---|---|---|---|---|---|---|---|---|
| CER | RMSE | 12.85 | 13.36 | 13.12 | **12.74** | 13.73 | 14.08 | 13.13 | 13.70 |
| | MAE | **8.32** | 8.81 | 8.64 | 8.35 | 8.77 | 8.93 | 8.48 | 8.83 |
| | MBE | -3.30 | -3.15 | -2.82 | -3.33 | -4.09 | -4.95 | **-2.34** | -3.76 |
| COT | RMSE | 8.96 | 9.87 | 9.55 | 9.07 | 7.91 | **7.71** | 7.84 | 7.86 |
| | MAE | 4.55 | 5.52 | 5.32 | 4.67 | 4.01 | 3.96 | **3.90** | 4.02 |
| | MBE | -0.89 | -0.26 | **0.06** | -0.96 | -0.53 | -0.26 | -0.91 | -0.53 |



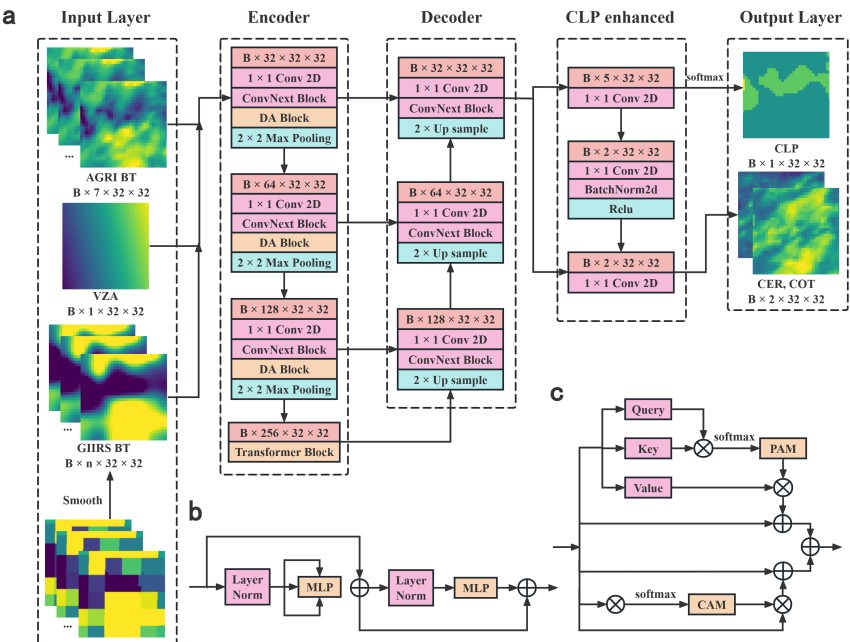

**Figure 1.** (a) Architecture of the HIR-COMP-Unet. (b) Transformer block. (c) Dual attention block, consist of position attention module (PAM) and channel attention module (CAM). Further details can be found in the supplementary materials.

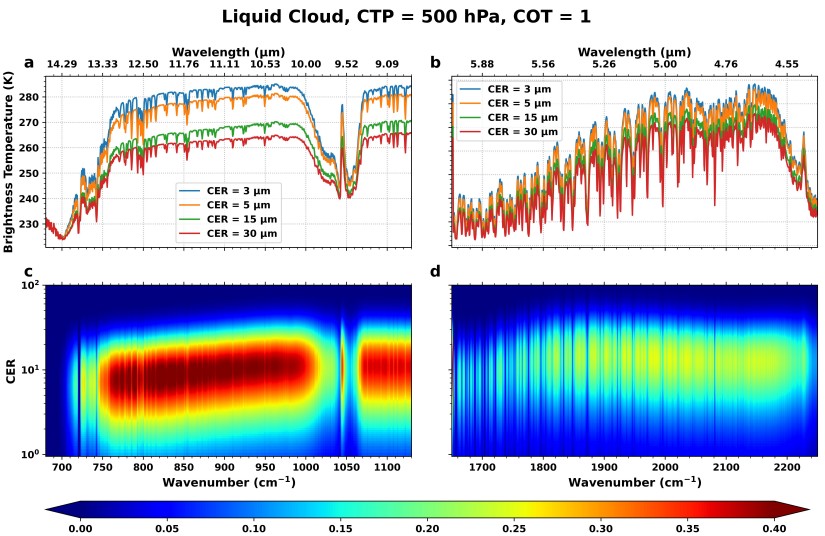

**Figure 2.** BT response of GIIRS channels to liquid cloud properties: (a) LWIR (680-1130 cm⁻¹) and (b) MWIR (1650-2250 cm⁻¹) spectral channels for varying CER; (c) LWIR and (d) MWIR sensitivity to CER variations.


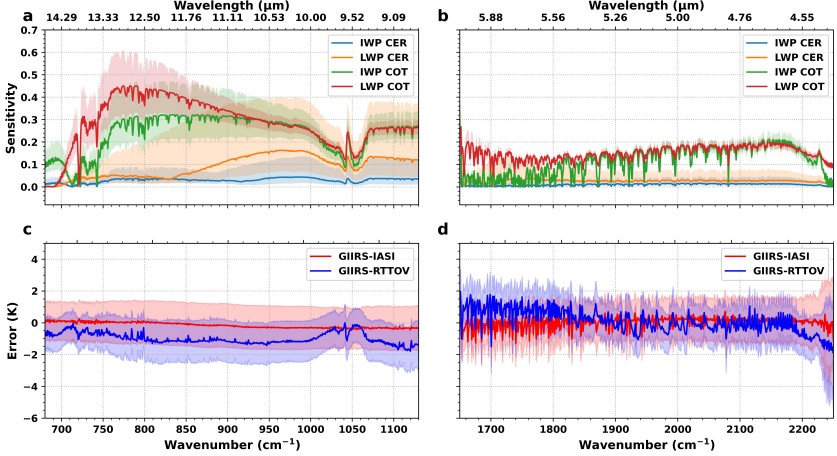

**Figure 3.** The average sensitivity of FY-4B/GIIRS LWIR (a) and MWIR (b) spectral channels to COMP variations. Bias (solid line) and STD (shaded area) of GIIRS LWIR (c) and MWIR (d) taking IASI BT (red) and RTTOV simulation (blue) as 850 baselines, respectively.





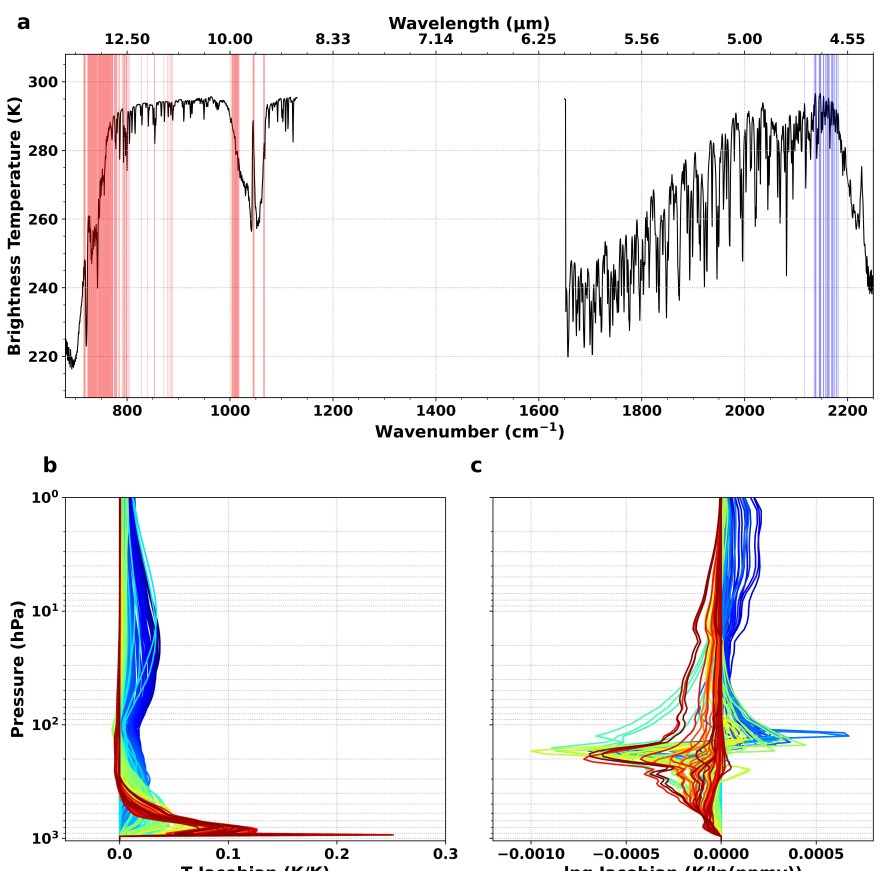

**Figure 4.** Channel selection of FY-4B/GIIRS LWIR (red vertical lines) and MWIR (blue vertical lines) after screening (a). The temperature Jacobian of channels (b), and the water vapor mixing ratio (lnq) Jacobians (c) of GIIRS channels we selected.





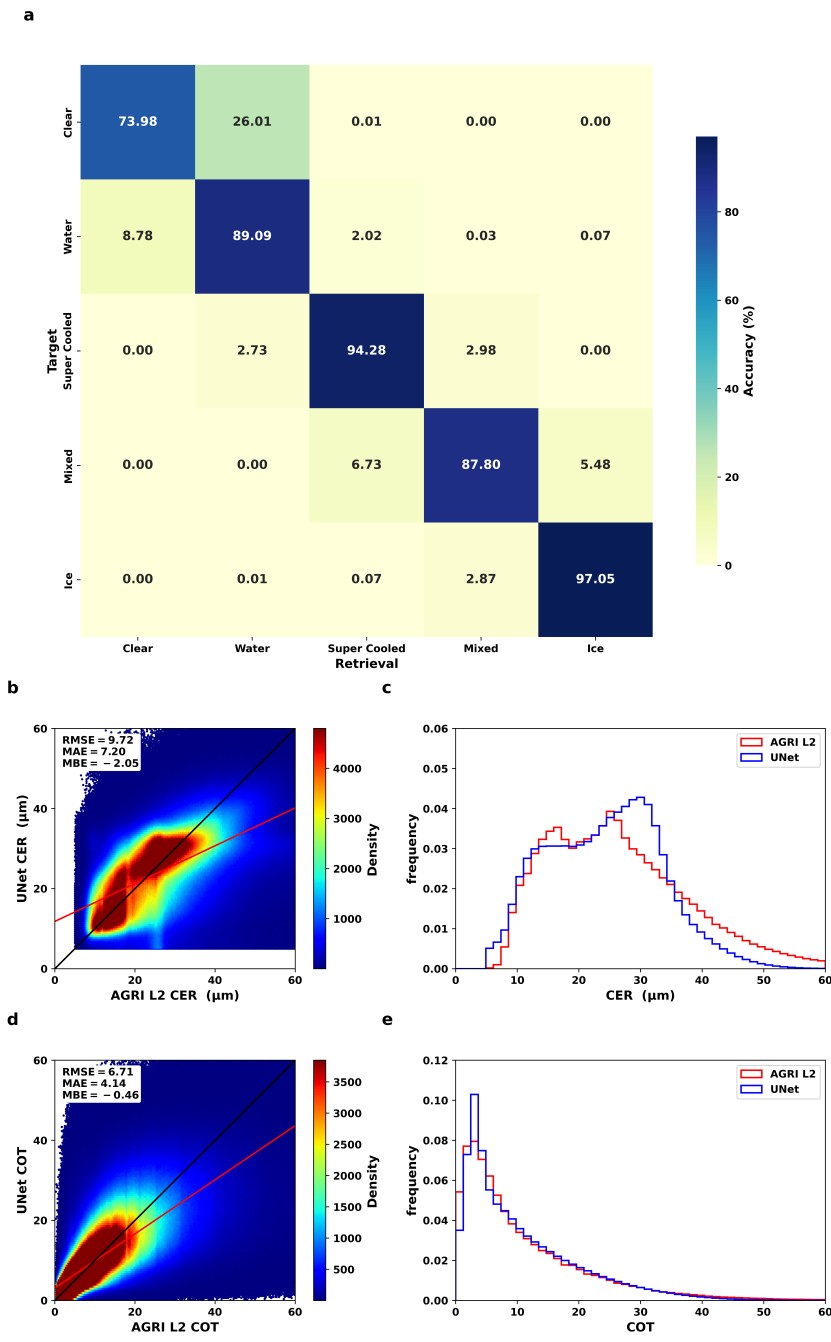

**Figure 5.** Independent validation of HIR-COMP-Unet (Unet) retrieval performance

using only AGRI channels compared to baseline AGRI L2 COMP products during



daytime conditions: (a) Confusion matrix for CLP identification showing classification accuracy; (b, d) Density scatter plots comparing retrieved versus reference CER and COT, with 1:1 line (black solid) and regression fit (red solid); (c, e) Probability density functions s of CER and COT.



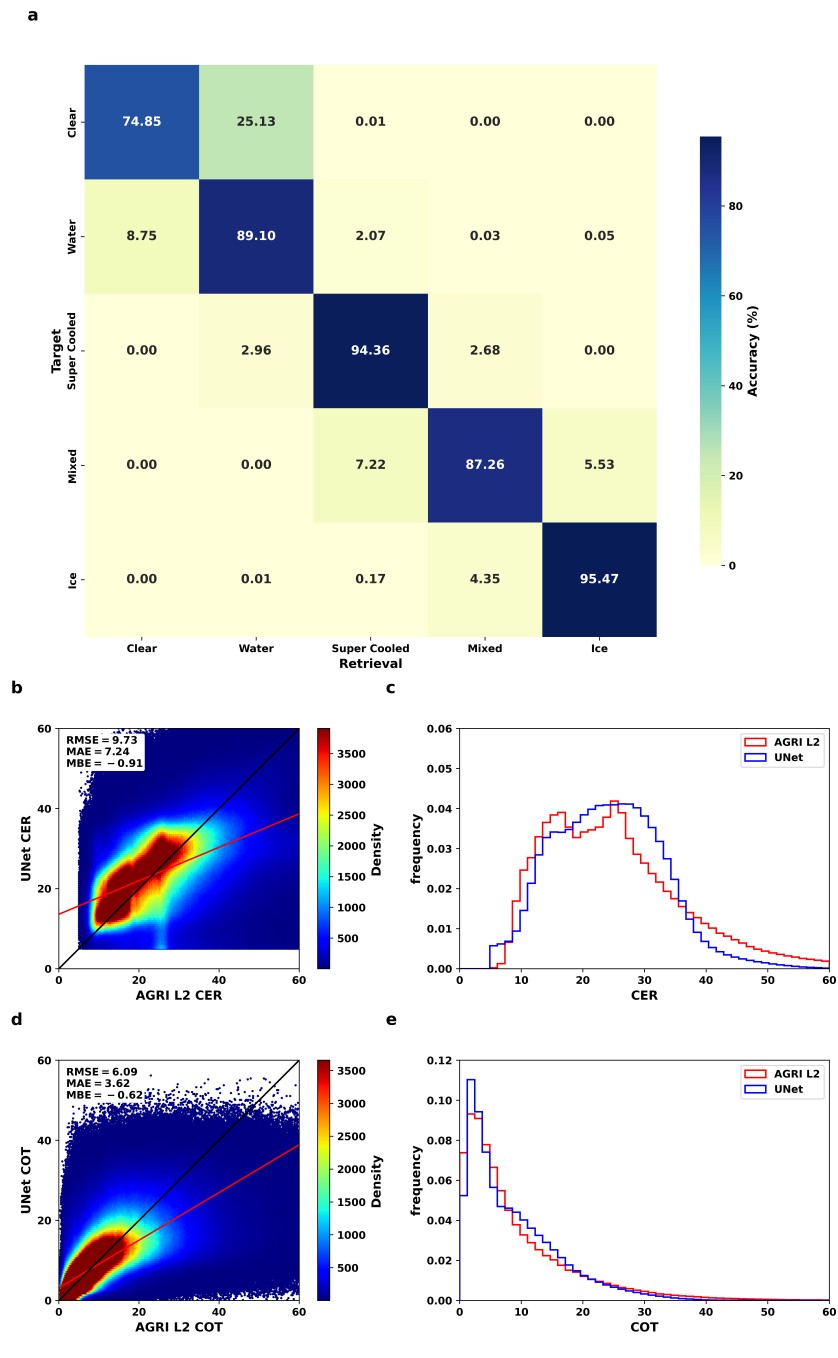


**Figure 6.** Independent validation of HIR-COMP-Unet retrieval performance using 30 selected GIIRS LWIR and 10 MWIR channels compared to baseline AGRI L2 COMP





products during daytime conditions: (a) Confusion matrix for CLP identification

showing classification accuracy; (b, d) Density scatter plots comparing retrieved

versus reference CER and COT, with 1:1 line (black solid) and regression fit (red

solid); (c, e) Probability density functions s of CER and COT.

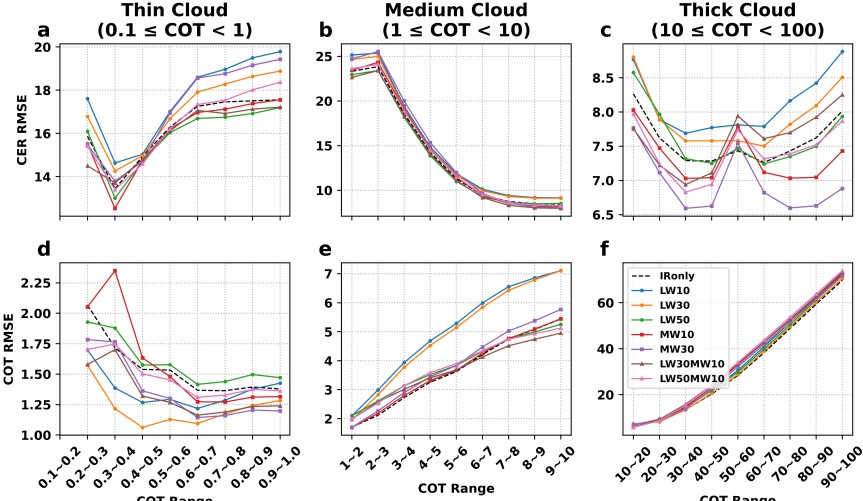

**Figure 7.** RMSE of CER (a, b, c) and COT (d, e, f) retrievals using varying numbers

of GIIRS channel inputs, stratified by COT values: thin (COT = 0.1-1), medium (COT

= 1-10), and thick (COT = 10-100) clouds.







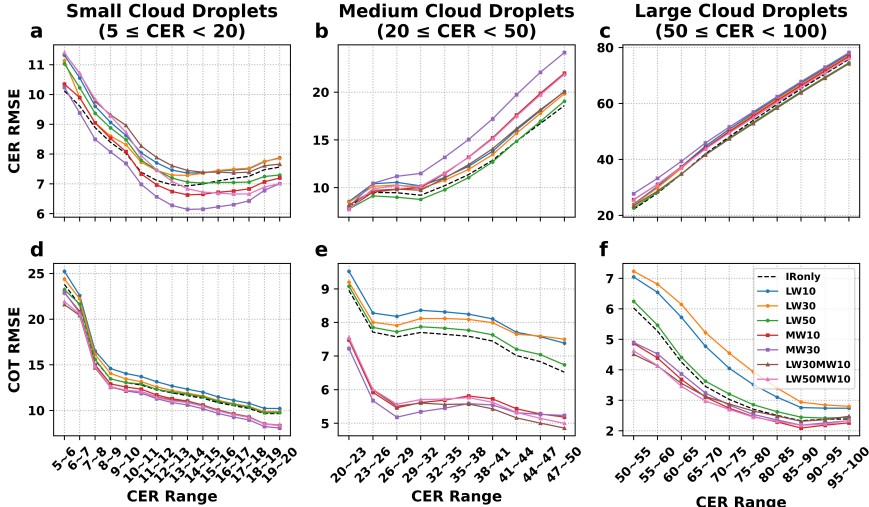

**Figure 8.** RMSE of CER (a, b, c) and COT (d, e, f) retrievals using varying numbers

of GIIRS channel inputs, stratified by CER values: Small (5-20), medium (20-50), and

thick (50-100) cloud droplets.




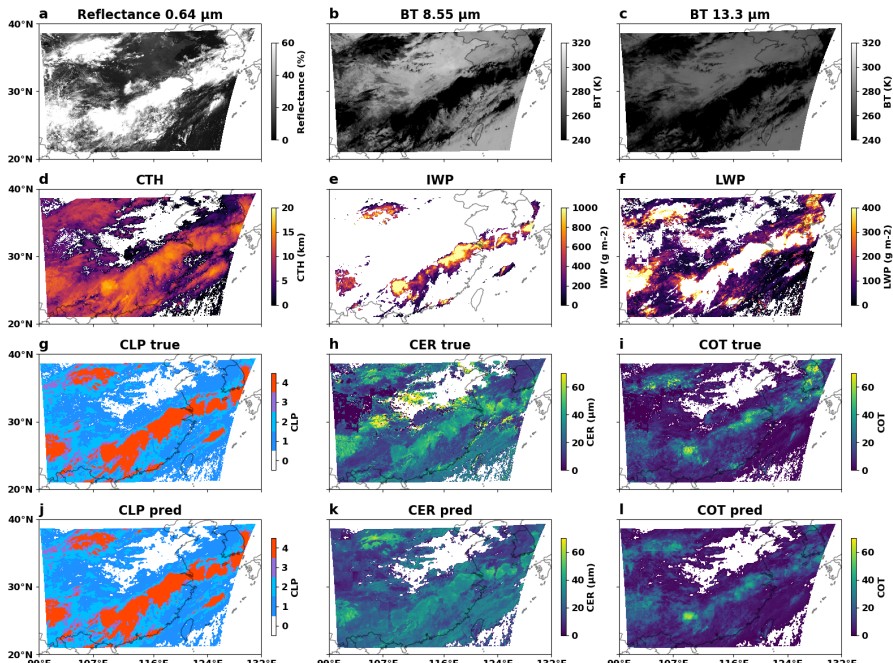

**Figure 9.** Comparative demonstration of cloud property retrievals between FY-4B/AGRI L2 products (true) and the HIR-COMP-Unet model retrievals (pred) for a daytime case (03:15 UTC, 30 June 2024). (a, b, c) show AGRI imagery for the 0.64 μm, 8.55 μm, and 13.3 μm channels respectively. (d, e, f) show the CTH, ice water path and cloud water path products from DCOMP. (g, h, i) show the CLP (0, 1, 2, 3, 4 represent clear, water, super cooled, mixed and ice phase cloud, respectively), CER and COT from AGRI L2 products. (j, k, l) show the CLP, CER and COT from HIR-COMP-Unet retrievals.



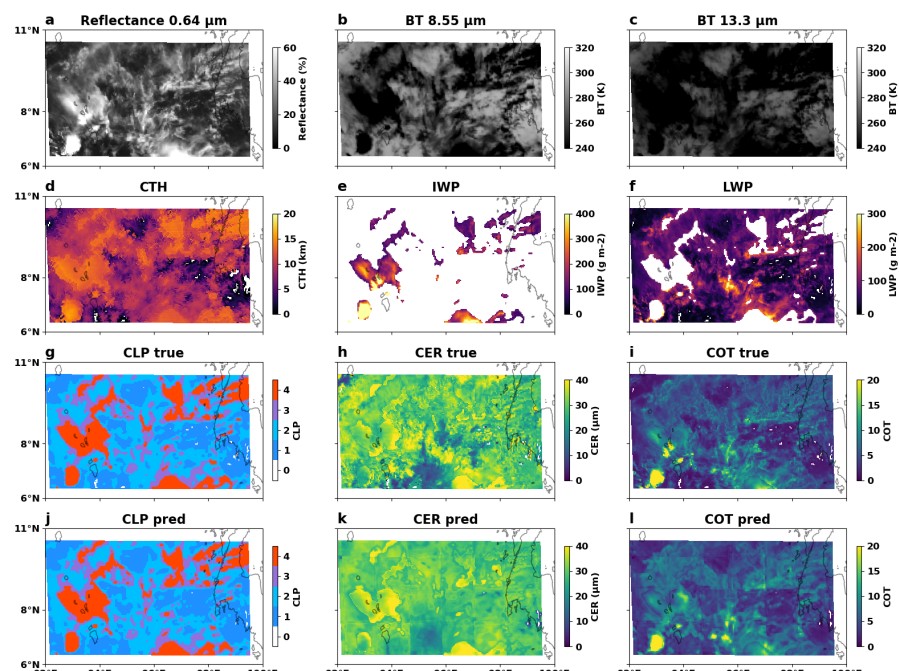

**Figure 10.** Same as Figure 9, but for another daytime case (04:15 UTC, 10 April 2024).



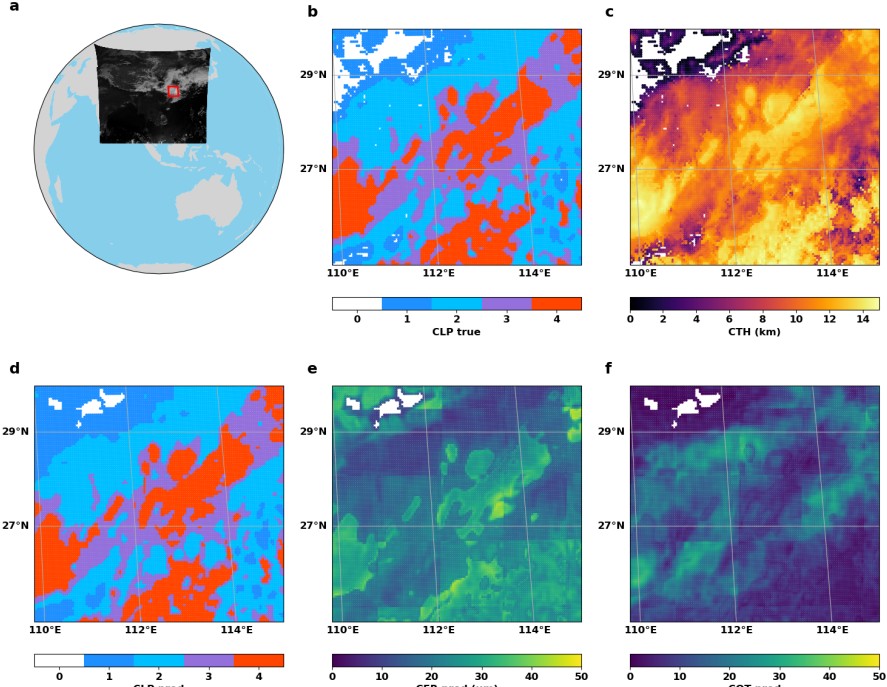

**Figure 11.** Nighttime cloud property retrievals from HIR-COMP-Unet over the study region (a) (centered at 27.5°N, 112°E) at 11:00 UTC on 29 June 2024. (b, c) AGRI L2 operational products showing cloud phase (CLP) and cloud top height (CTH), and (d, e, f) HIR-COMP-Unet retrievals of CLP, CER, and COT.



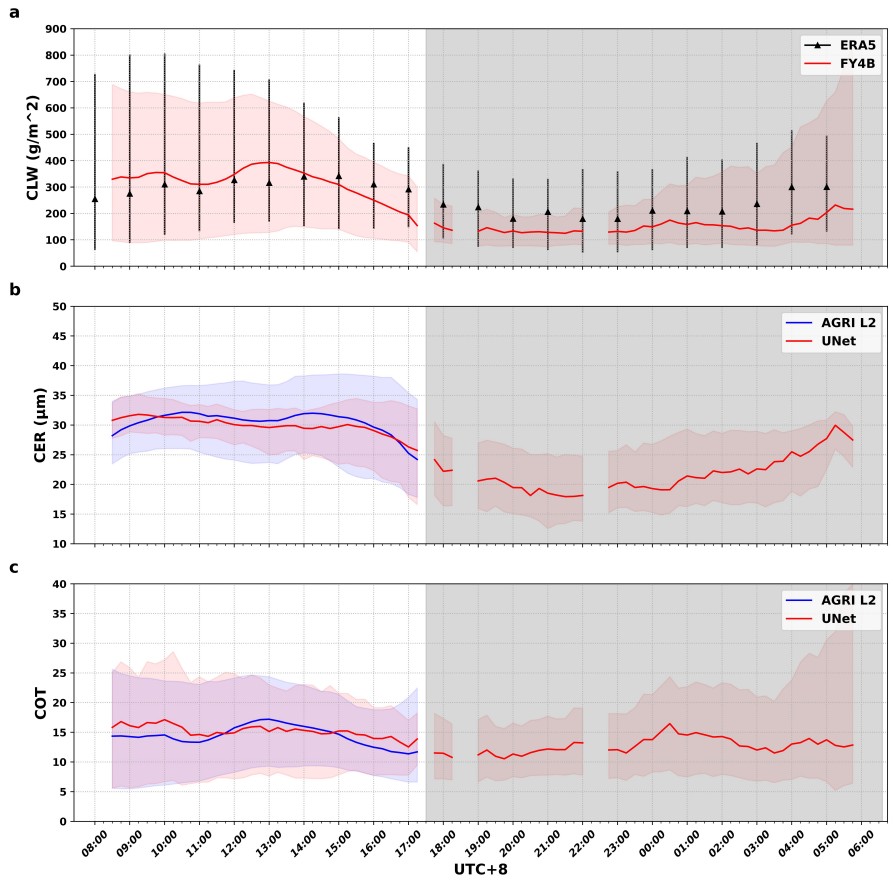

925

**Figure 12.** Diurnal cycle of cloud water path (CWP), CER, and COT distributions over the study region (see Figure 11) during 29 June 2024: (a) Comparison of CLW quantiles (25th, median, 75th) between ERA5 reanalysis and HIR-COMP-Unet model retrievals; (b) CER and (c) COT quantile comparisons between operational FY-4B/AGRI L2 products and HIR-COMP-Unet model. Gray shading indicates nighttime periods (UTC+8), during which all retrievals derive from HIR-COMP-Unet due to the limitations of DCOMP operational algorithm.