# Peer review of "Enhancing nighttime cloud optical and microphysical properties retrieval using combined imager and sounder from geostationary satellite"

_EGUsphere, 2025_

## Referee Comment (RC2)

**egusphere-2025-2928 for AMT**

"Enhancing nighttime cloud optical and microphysical properties retrieval using combined imager and sounder from geostationary satellite" (Xia et al.)

This study details a machine learning framework (Dual-Attention Transformer U-net) that improves the accuracy of retrieving nighttime cloud properties by integrating data from a hyperspectral infrared sounder (GIIRS) and a high-resolution imager (AGRI) on a geostationary satellite. It would be a good attempt to add HIR sounder data to improve the accuracy of nighttime cloud microphysics products from passive imagers, which has long suffered due to the lack of visible band information at night. However, I would think that several major points should be further complemented for its publication. Please see below for several questions/suggestions.

1) It appears that the addition of HIR may not substantially improve the nighttime retrieval of cloud microphysical products, despite the advanced AI/ML skills adopted in this study. I remain uncertain about the effectiveness of IR channel information from the HIR sounder for retrieving cloud microphysical properties, given the inherent limitations of such channels in capturing particularly these cloud microphysics quantities. I agree that HIR data could be more beneficial for other cloud retrievals, such as cloud height, phase, or temperature/humidity profiling. I would recommend that the authors provide a more detailed explanation of their underlying hypothesis and the physical rationale for expecting improvements in cloud microphysical retrievals when incorporating HIR data with the Dual-Attention Transformer U-net approach.

2) The main focus of the study is on nighttime improvement. However, it seems that the current manuscript does not provide sufficient evaluation results beyond Figure 11. I understand that the available data resources are quite limited for this, as the authors have already demonstrated through the use of DCOMP comparisons. But such diurnal cycles wouldn't be proper 'truth' data at night due to NCOMP's natural shortcoming even if the products are operational as the authors already indicated in 3.4. I would like to suggest considering VIIRS Day/Night Band–incorporated products from JPSS satellites (Suomi-NPP, NOAA-20, and NOAA-21), which are publicly available (for example, on the Amazon Web Services server), as a potentially valuable additional resource for nighttime cases.

3) As this study uses both imager and sounder which have very different resolutions. More details for the spatial matchups for the HIR-COMP-Unet model training will be desirable in the manuscript.

Minor comments:

There are several errors in citing papers. Please double check them:

Line 80 and others: Correct Andi et al. 2013 to Walther et al. 2013. The first reference info should be corrected as well.

Line 123: Charles et al. (2024) -> White et al. (2025).

Line 150: Guo et al. (2024) missing in the references

Just a suggestion: to reveal more effectively the main work of this study, please consider adding "with machine learning" to the title

Abstract:

Line 35: the same geostationary (GEO) platform with a machine learning (ML) framework.

Line 40: It is important to provide the main satellite information here. Please add something like "(COT) when applied to FY-4B AGRI and GIIRS" in the abstract.

Line 45: Middle -> Mid

Line 70-72: Need the full names for VIRS/NIR and SWIR

Line 92: TIR only will be enough. It's already defined above.

Line 134 and others: Since this study uses both imager and sounder, please consider including micrometer units together in the brackets throughout the manuscript. I understand in several places, the authors mix-use wavenumbers (cm-1) and wavelengths (micrometer), which will be intentional for hyperspectral sounders and imagers, respectively, but it would be helpful for better understanding for readers and for some consistency.

Line 143: add on board FY-4 satellite series or specify the one after "(GIIRS)"

Line 155: Remote (Guo et al., 2024) to avoid repetition

Line 234: Please add a little bit more details about "CLP enhanced layer".

Line 237: gaussian -> Gaussian

Line 253: A little more details about 'CLP feature maps' and its expected role in this model will be helpful. Is this about 'CLP-enhanced' layer?

Line 547: Please double-check the error values, 9.73 or 9.72.